# Development of Solar Powered Biodiesel Reactor for Kuwait Sheep Tallow

**Fnyees Alajmi [1,\*], Amer Alajmi [1], Ahmed Alrashidi [1], Naser Alrashidi [1], Nor Mariah Adam [2] and Abdul Aziz Hairuddin [2]**

1   Public Authority for Applied Education and Training, Adailiya 00965, Kuwait; amerq880@gmail.com (A.A.); ahmedalrashidi2017@gmail.com (A.A.); naser.alrashidi.upm@gmail.com (N.A.)
2   Department of Mechanical and Manufacturing Engineering, Universiti Putra Malaysia, Serdang 43400, Selangor, Malaysia; mariah@eng.upm.edu.my (N.M.A.); ahziz@upm.edu.my (A.A.H.)
\*   Correspondence: alktab@hotmail.com

**Abstract:** Biodiesel is one of the more recent green fuel products in the world. It can be produced from several raw materials such as straight vegetable oils, animal fats, tallow, and waste cooking oils, and blended with diesel. Properties of biodiesel are different compared to fossil diesel in terms of production methods and emission levels released after combustion in an internal combustion engine. Kuwait consumes a huge amount of energy which is almost 8% to meet the increasing demand for electricity and water. Moreover, the use of electricity in the production of biodiesel increases energy use and cost of production. Kuwait is receiving an amount of solar irradiation ranging from 2050 kWh/m$^2$ to 2100 kWh/m$^2$. The present study is concerned with the evaluation of the potential to use solar energy to produce biodiesel from sheep fat waste as a raw material. An experimental test rig was set up for a single cylinder diesel engine in the laboratory, where a solar power system was used to assist the production process of biodiesel from tallow waste. The biodiesel is then blended with diesel at different volume percentages, such as graded as B20, B50, B75 and B100, respectively. The exhaust gases such as oxygen, carbon monoxide, carbon dioxide, nitric oxide and nitric dioxide where also analyzed. An optimum decrease in values of nitric oxide levels was observed at the load of 51%, 68%, 85% and 93% during the operation at blend of biodiesel B20, B50, B75 and B100, respectively. Nitric dioxide was decreased at the load of 51%, 68% and 85% during the operation using B20, B50 and B75, respectively. Optimum SFC was achieved at B20, B50 and B75 during high loads of 85% and 93%. It can be concluded that sheep tallow biodiesel shows a promising result in terms of fuel consumption and environmental emissions of greenhouse gases.

**Keywords:** solar power; biodiesel; Kuwait; sheep tallow

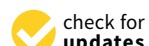



## 1. Introduction

Energy is a very important factor for human endeavors. It is needed for economic growth and basic human needs such as food and health [1]. In recent years, the world is faced with energy crisis due to increased population growth, increased consumption of energy and the depletion of energy resources [2]. According to Abbas and Othman [3], the energy consumption of the world doubled from 256 to 505 million GJ between 1973 and 2007. This crisis has expanded with a further decline in world petroleum reserves, leading to a reduction in oil production. In fact, at the current rate of energy consumption, the crude oil reserves are estimated to be completely depleted in the next 44 years [4]. Furthermore, most of the world's energy sources are from coal, petrochemical and natural gases [5]. These sources have been reported to be finite and will be depleted shortly due to the current rate of consumption [6]. These situations have led to the search for environmental and sustainable sources of energy.

Sustainable energy sources such as solar, biomass, geothermal, wind and hydro are renewable and readily available, and can also be obtained at an affordable cost with

less impact on the environment. These energy sources can adversely lead to long-term sustainable development [7]. Biomass is one of the most common and important renewable sources of energy that can be obtained from wood, animal waste, plants and municipal waste. Biomass can be directly burnt or easily processed into biofuels such as methane (biogas), ethanol (bioethanol) and biodiesel, of which biodiesel is the most common and affordable biofuels [3,6]. Therefore, there is a strong need to replace fossil fuels with more sustainable and readily available renewable energy such as biodiesel.

Biodiesel is a renewable source of energy produced from the reaction between biomass materials (vegetable oil, animal waste oil) and alcohol, often carried out in the presence of a catalyst. Biodiesel is a liquid fuel commonly known as B100 or neat biodiesel in its pure and unblended form. Biodiesel has been in existence since 1893, when Dr. Rudolf Diesel developed the first vegetable oil fueled engine [8]. However, the full exploration of biodiesel based on vegetable oil became of significant interest only in the 1980s, due to the increasing demand for a renewable and sustainable energy source that will also reduce greenhouse gas (GHG) emissions [9,10]. Since then, biodiesel has slowly penetrated the market in Europe, especially in Germany and France, as an alternative to petrol diesel [11,12]. Commercially, these biodiesel blends are named B5, B20 or B100 to represent the volume percentage of biodiesel component in the blend with diesel as 5, 20 and 100 percentage volume, respectively. Currently, many countries around the world have explored and commercially used biodiesel blends for their vehicles such as the United States, Japan, Brazil and India [9,13]. However, most countries, especially in the Middle East, still rely heavily on fossil fuel for its energy demand.

Countries in the Middle East such as Kuwait consume a huge amount of energy to meet the increasing demand for electricity and water [14]. Al-Nassar et al. [14] reported that the energy consumption rate of the state of Kuwait is at 8% annually. The rising rate in the total energy consumption is largely driven by the increased demand from power stations and water desalination plants [15]. There is also a rise in the use of diesel generators, especially during summer at the peak of air conditioning demand. Increases in the country's fuel production by increasing the capacity of available refineries and building of new ones have failed to meet the energy requirements of the country. Meanwhile, the electricity demand continues to grow at 5% per year. Kuwait's oil consumption is likely to continue increasing, owing to population growth and urbanization, the growth of motor car ownership and rising living standards, all of which are tied to economic growth related to the oil industry and the rise in world oil prices. Thus, there is a need to urgently diversify the energy sector by adequate development of renewable energy sources as a solution to Kuwait's lingering energy problems. Therefore, this study investigated the development of solar energy powered biodiesel production using animal fat wastes from sheep tallow to analyze and characterize the properties of biodiesel produced from sheep tallow and to determine the effect of solar energy on the performance of a biodiesel reactor.

## 2. Materials and Methods

### 2.1. Experimental Procedures

In this study, sheep fat waste was collected from abattoir and transesterified to produce biodiesel. A heterogenous transesterification process was used in this research. Potassium hydroxide (KOH, 86%, Sigma-Aldrich, Louis, MO, USA), magnesium oxide (MgO, Sigma-Aldrich, 97%) and ammonium carbonate ($(NH_4)_2 CO_3$, Sigma-Aldrich, 99%) were used for the transesterification process. Animal fat from sheep was used as feedstock in the production of biodiesel. The animal fats were extracted from sheep prior to the production of biodiesel. The production of the biodiesel fuel also involved the design and fabrication of a solar powered biodiesel reactor. Details of the experimental procedures are outlined in the following sections. Figure 1 illustrates the schematic diagram of the engine test ring.

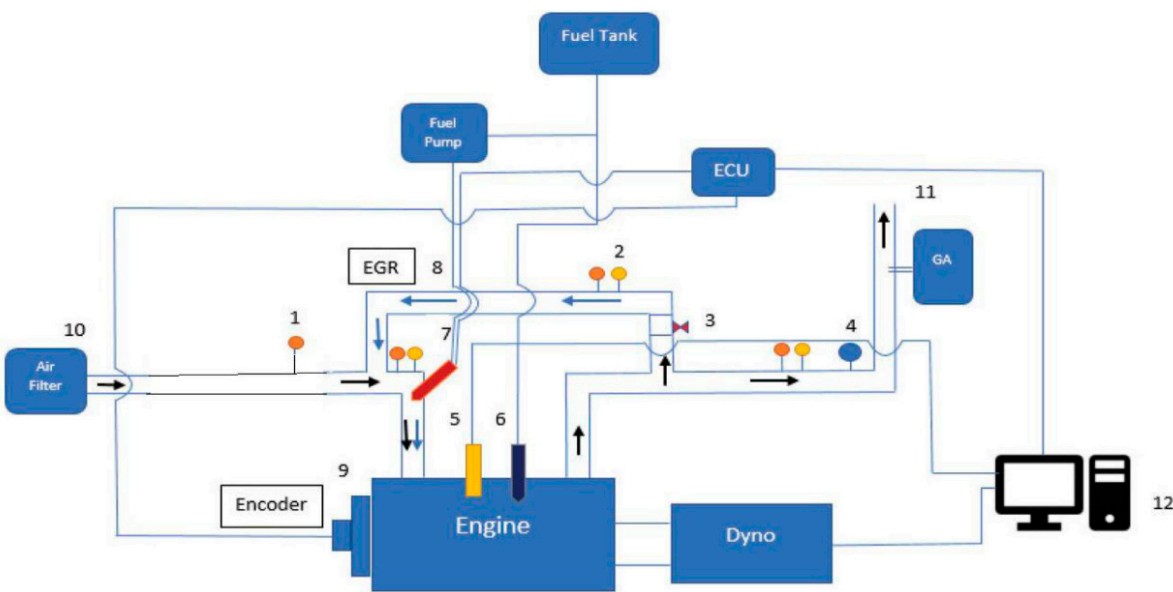

**Figure 1.** Schismatic diagram of engine test rig: (1) Thermocouple, (2) Pressure sensor, (3) EGR valve, (4) Oxygen sensor, (5) In-cylinder pressure, (6) Direct injection, (7) Port injection, (8) Exhaust gas recirculated system, (9) Encoder, (10) Opening intake air, (11) Opening exhaust air, (12) PC.

We used two main processes of solar-assisted biodiesel equipment design and fabrication, and our evaluation of equipment performance was in terms of biodiesel performance in the engine. The engine emissions and fuel properties were studied. A commercial computer aided design and drafting software called AutoCAD version 19.1 (Autodesk Inc., Mill Valley, CA, USA) was employed for the design of the solar assisted biodiesel plant. Details of the plant design is shown in Figure 1. Super Pro design software (Intelligent, Inc., Scotch Plains, NJ, USA) was also used to determine the material and energy balance.

Transesterification is a chemical process of mixing fat or oil with an alcohol to produce esters and glycerol. These reactions often take place in the presence of an acid or base catalyst. It transforms the large, structured bio-oils into smaller, straight-chain molecules, with fuel characteristics like those of fossil fuel. Methanol is the most used alcohol because of its low price compared to other alcohols. The transesterification reaction of a triglyceride with methanol is presented by a chemical reaction as shown in Figure 2.

**Figure 2.** General representation of the catalytic transesterification process of triglycerides with methanol to produce biodiesel.

## 2.2. Extraction of Fat from Sheep

For our research, 4 kg to 10 kg net weight of sheep tail were purchased from slaughterhouse, according to the age group (determined by fathel) of the Department of Statistics, Kuwait. An experiment was performed to determine the ratio of the liquid to the solids. Figure 3a shows the raw sheep fat from slaughterhouse. The amount of 1000 g of sheep fat was cut and minced into small pieces (Figure 3b). The minced fat was further used as raw material for biodiesel production. The fat was then placed on a heating source of 150 °C for 20 min until the liquid was distilled. After that, the crude oil is then filtered from the solid impurities by metal filters. After the oil temperature drops to 70 °C, the oil is filtered more accurately using cotton filters, as in Figure 3c. The extraction yield was estimated to be 19.1%.

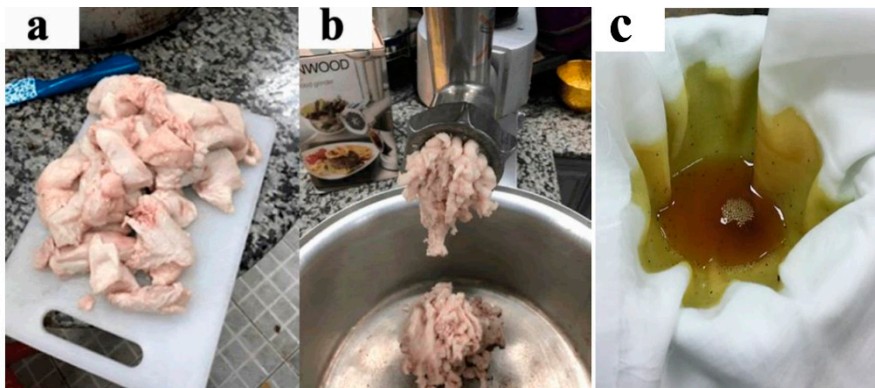

**Figure 3.** (**a**) Sheep fat chopped; (**b**) the fat minced to small pieces by mincing machine, (**c**) the process of fine filtration of crude oil.

## 2.3. Solar Power System Design and Calculations

Solar energy concepts have been widely applied for water heating systems as well as electricity production solar power systems. As highlighted in the review section, the reaction temperature to produce biodiesel is achieved using electricity, which significantly increases production costs. Even though some recent studies have developed solar assisted biodiesel reactors, there is no known study on the use of flat-plate solar panels for powering an animal fat waste biodiesel plant. In this study, a flat-plat solar collector was installed to power the biodiesel production plant. Just like in other solar collectors, temperature plays a vital role and has an effect on the efficiency and maximum photovoltaic output of the flat-plate solar panel. The factors affecting the yield potential of the panel are the ambient temperature and the type of installation of the temperature coefficient of the actual panel. A typical flat-plate solar power system consists of a photovoltaic collector system, battery bank, inverter and control unit. The geographical location of the biodiesel lab is 29.3798 latitude and 47.9878 longitude. Figure 4 shows the location and the amount of irradiation from 2050 kWh/m$^2$ to 2100 kWh/m$^2$; this value is efficient for solar powered biodiesel.

A solar photovoltaic system (SPS) includes different components that were selected according to application types, and the necessity to produce power at appropriate condition. The various SPS components include a PV collector, bank of batteries, solar charge controller, inverter and loads, as shown in Figure 5a. The photovoltaic panel in this study was made from semiconductors such as monocrystalline and silicon, and the panel converts the sunlight to DC electricity (Figure 5b). Energy supplied from PV panels is stored in bank of batteries (Figure 5b) which allows the system to be used during the day and night or unstable weather. The battery is made from lead acid and can discharge or recharge around 80% of its capacity. This device regulates voltage and currents in the solar power system and is also called the controller (Figure 5c). It basically regulates charging from PV panels to batteries and cuts the charging if the battery bank is full of storage. The inverter is the

part of solar power system that converts the DC output from PV panels to the AC current (Figure 5d). Inverters regulate the output power and installation type. The efficiency of all inverters is around 90% when the load demand is greater than 50% of the rated load.

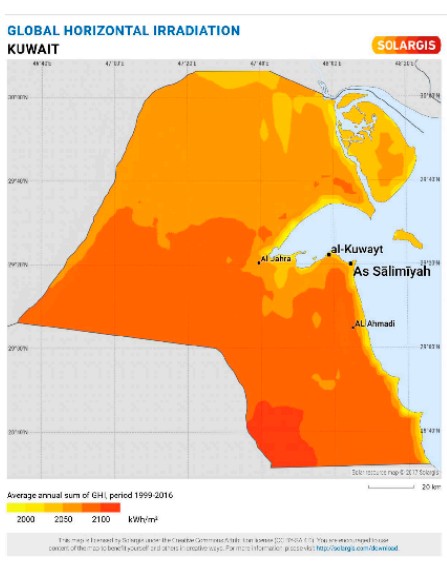

**Figure 4.** Horizontal irradiation distribution in Kuwait.

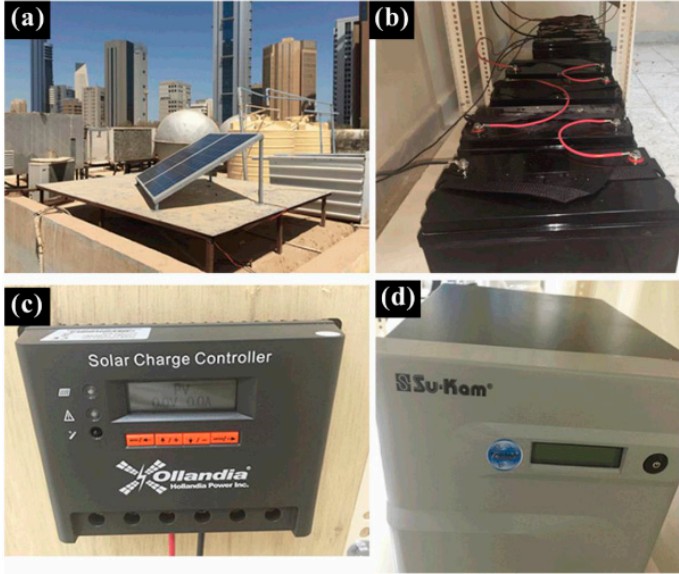

**Figure 5.** Solar power system set-up for the biodiesel production; (**a**) solar panel over roof surface; (**b**) solar power system 24-volt batteries; (**c**) solar charge controller; (**d**) solar AC/DC converter.

a      PV Collector

The photovoltaic panel in this study was made from semiconductors such as monocrystalline and silicon; the panel converts sunlight to DC electricity (Figure 6).

b      Bank of Batteries

Energy supply from the PV panels is stored in the bank of batteries (Figure 6) which allows the system to be used during the day and night or unstable weather. The battery is made from lead acid and can discharge or recharge around 80% of its capacity.

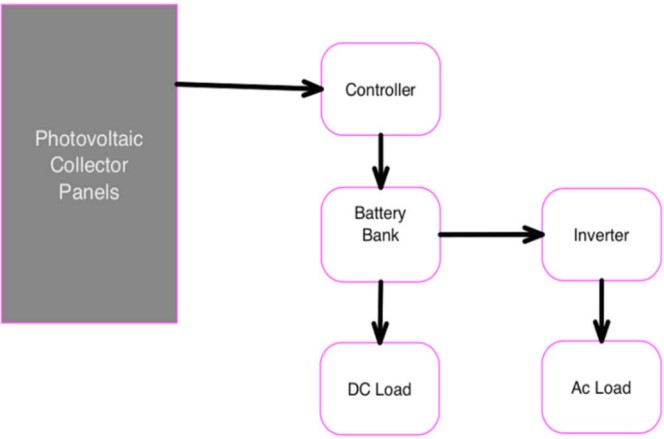

**Figure 6.** Schematic diagram for a stand-alone solar power system.

c    Solar charge controller

This device regulates voltage and currents in the solar power system and is also called the controller (Figure 6). It basically regulates charging from PV panels to batteries and cuts the charging if the battery bank is full of storage.

d    Inverter

The inverter is the part of the solar power system that converts the DC output from the PV panels to AC currents (Figure 6). The inverter regulates the output power and installation type. The efficiency of all inverters is around 90% when the load demand is greater than 50% of the rated load.

e    Loads

The electric loads in watts connect to the solar power system, and they can be motors, solenoid valves, switches, pumps, lights, etc. The electric devices used in this study includes a pump, solenoid valves, an agitator and an electric heater. Table 1 explains the various devices and their rating.

**Table 1.** Solar power system's devices and ratings.

| Individual Loads | Qty | Volts | AC Amps | Watt AC | Use h/d | Use d/w | /7 days | W.h.AC |
|---|---|---|---|---|---|---|---|---|
| Agitator | 1 | 220 | 0.45 | 80 | 4 | 7 | 7 | 320 |
| Solenoid valve | 3 | 220 | 0.25 | 150 | 3 | 7 | 7 | 450 |
| Pump | 1 | 220 | 0.4 | 70 | 5 | 7 | 7 | 280 |
| Heater | 1 | 220 | 3 | 1500 | 0.5 | 7 | 7 | 750 |
| AC Total Connected Watts | | | | 1800 | AC Average Daily Loads | | | 1800 |

Overall, the 8 batteries installed in 4 units were connected in series. Each unit contains two connected batteries, with a 48-volt parallel module in line with the converter specifications. The installed solar power system was capable of securing an electric source of 1000 watts for 8 h or 2000 watts for 4 h. The feedstock used for the biodiesel production process is sheep tallow.

*2.4. Diesel Engine Instrumentations*

The power of the engine was measured by using electrical load gradually. Loads consist of specified electric heater (3000 watts) that can be adjustable from light load to high load. The reading of the engine starts from both side with and without loads. Two thermocouples were used to measure the temperature of air entering the diesel engine intake and the exhaust gas temperature. The thermocouple for intake air temperature was

installed directly before the intake port of the throttle body valve, while the thermocouple used to measure the temperature of exhaust gases was installed directly after the exhaust port to avoid any heat losses to the environment. Both thermocouples were k-type with a common range of −270 °C and 1250 °C. An air flow mass sensor was used to measure the air mass flow enters the engine. The device consisted of two-part AAV3 (Figure 7a) and DL2 (Figure 7b) data loggers for air mass flow sensor (Fieldpiece Instruments, Inc., Orange, CA, USA). The air flow mass sensor was installed and fixed to the diesel engine intake port. An Exhaust Gas Analyzer device was used to measure the proportion of combustion exhaust gases, differential flue temperatures, carbon dioxide, flue and ambient carbon monoxide and emission of greenhouse gases. The Exhaust Gas Analyzer used in this experiment is Eagle x155 with specifications shown in Table 2. In this experiment, a simple graduated cylinder (Figure 7c,d) was used for measuring the fuel consumption. This graduated cylinder can clearly show the consumption of both fossil diesel and biodiesel with high accuracy volume reading.

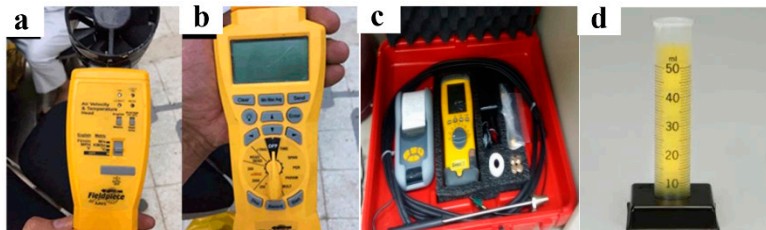

**Figure 7.** (**a**) Fieldpiece AAV3 air mass flow sensor; (**b**) DL2 data logger, (**c**) Exhaust Gas Analyzer (Eagle x155), (**d**) graduated cylinder.

**Table 2.** Combustion analyzer specification.

| | |
|---|---|
| **Temp Measurement** | |
| Flue Temp Range | 20–2400 °F/−29–1315 °C |
| Inlet Temp (robe-T2) | 20–2400 °F/−29–1315 °C |
| Inlet Temp Range (Ambient) | 2–112 °F/0–50 °C |
| Net Temp (NET) | 20–2400 °F/−29–1315 °C |
| Resolution | 0.1 °F/°C |
| Flue (T1, Inlet T2, NET) Accuracy | ±[0.3% rdg +3.6°F(2 °C)] |
| Flue (T1, Inlet T2, NET) Accuracy | ±[0.3% rdg +3.6°F(2 °C)] |
| Inlet Temp Accuracy | ±[0.3% rdg +1.8°F(1 °C)] |
| Gas Measurement | |
| Oxygen | 0–21% |
| $O_2$ Resolution/Accuracy | 0.1%/±0.3% |
| Carbon Monoxide | 0–2000 ppm (4000 max 15 min) |
| CO Resolution/Accuracy | 1 ppm/±10 ppm <100 ppm, ±5% rdg >100 ppm |
| Carbon Dioxide | 0–20% |
| $CO_2$ Resolution/Accuracy | 1.0%/±0.2% |
| Efficiency | 0–99.9% |
| Efficiency Resolution/Accuracy | 0.1%/±3% |
| Excess Air | 0–250% |
| Excess Air Resolution/Accuracy | 0.1%/±3% reading |
| $CO/CO_2$ ratio | 0–0.999 |
| $CO/CO_2$ resolution/accuracy | 0.001/± 5% rdg (reading) |
| Pressure | |
| Range: | Accuracy |
| ±0.08″ wc (±0.2 mBar) | 2 ± 0.002″ wc (±0.005 mBar) |
| ±0.4″ wc (±1 mBar) | ±0.01″ wc (±0.03 mBar) |
| ±32″ wc (±80 mBar) | ±3% rdg |
| | 0.001″ wc < 9.999″ wc |
| | 0.01″ wc > 10.00″ wc |
| Resolution | 0.001 mBar < 24.999 mBar |
| | 0.01 mBar > 25 mBar |
| Pre-Programmed Fuels: Natural Gas, Propane, Heavy Oil, Light Oil, Bio Fuel, Wood | |

### 2.5. Exhaust Emissions Measurement

The measurement of the exhaust gases was conducted using a gas analyzer (NOVA 7466K) for measuring CO, $CO_2$, NO, $NO_2$ and $O_2$ emissions, as depicted in Figure 8. This was carried out to ensure the highest accuracy in the measurement, as this gas analyzer has different accuracy levels for different exhaust gases. NO and $NO_2$ sensors both have a resolution of 1 ppm. The $CO_2$ sensor has a resolution of 0.1 mol% of gas analyzed. At each test condition, data of different performance parameters and emissions were taken three times, and the averages were used to plot performance and emission graphs.

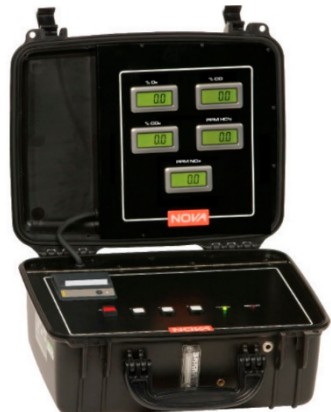

**Figure 8.** NOVA model 7466K used to measure the different emissions.

### 2.6. Calculation Methods

The brake-specific fuel consumption (*BSFC*) is defined as the ratio of the fuel consumption ($m_{fuel}$) to the brake power (*Bp*), as shown in Equation (1). The brake mean effective pressure (*BMEP*) is illustrated in Equation (2).

$$BSFC = \frac{m_{fuel}}{Bp} \tag{1}$$

$$BMEP = \frac{Torque \times 75.4}{Displacement \times PPR} \tag{2}$$

where LHV is lower heating value (kJ/kg) and *PPR* is power pulses per revolution.

## 3. Results and Discussion

### 3.1. Characterization of Sheep Fat Biodiesel and Blends

The properties of fuel types used in this study, including diesel, biodiesel blends, B20, B50, B75 and B100, were tested and analyzed by the Petroleum Research Centre, Kuwait Institute for Scientific Research. In this study, ASTM standards were employed to determine each property of the different fuels used. Table 3 provides the summary measurement properties of diesel and biodiesel blends, B20, B50, B75 and B100. To characterize pure sheep fat biodiesel (B100), several properties such as specific gravity, density, kinematic viscosity, total acid number, water content, total sulfur, flash point, lubricity, cloud points and pour points were examined and compared based on ASTM D6751 standards. Table 3 shows the detailed physicochemical characteristics of sheep fat biodiesel (B100) and its blends (B20, B50 and B75).

In Table 3, the kinematic viscosity values of diesel and biodiesel fuels were within the range as reported by previous research. The diesel fuel exhibited a lower value (3.5819 cSt) compared to the biodiesel fuel types. Meanwhile, B100 represented the highest value of kinematic viscosity relative to other fuels with (4.3390 cSt). For the density, B100 was found to be the highest (0.8649 g/cm$^3$), which exceeds the petroleum diesel (0.8339 g/cm$^3$), as shown in Table 3. For cloud point and pour point, results in Table 3 demonstrate that biodiesel blends increased with increasing content of diesel. Table 3 also presents the values of flash point for all the fuel types. Flash point is described as the temperature at

which a fuel gives off sufficient vapor to ignite in air. For total acid number, the biodiesel blends increased with increasing content of diesel. The values of sulfur content show that biodiesel has the lowest values as compared to fossil fuels, as can be seen in Table 3. Similarly, biodiesel fuels have the highest amount of water content compared to diesel.

**Table 3.** Properties of the diesel and biodiesel blends.

| Properties | Standards | Units | Diesel | B20 Diesel + Biodiesel 80:20 | B50 Diesel + Biodiesel 50:50 | B75 Diesel + Biodiesel 25:75 | B100 Biodiesel |
|---|---|---|---|---|---|---|---|
| Kinematic viscosity @ 40 °C | ASTM D445 | cSt | 3.5819 | 3.7333 | 3.9604 | 4.1497 | 4.3390 |
| Density @ 25 °C | ASTM D4052 | g/cm$^3$ °C | 0.8339 | 0.8401 | 0.8494 | 0.8572 | 0.8649 |
| Cloud point | ASTM D5773 | °C | −20 | −6.9 | −3.5 | 7.8 | 11.2 |
| Pour point | ASTM D5949 | °C | −43 | −25 | −12 | 2 | 15 |
| Flash point | ASTM D93 | °C | 90 | N/A | 89 | 87 | 97 |
| Total acid number | ASTM D664 | mgKOH/gm | 0.0110 | 0.0295 | 0.0572 | 0.0804 | 0.1035 |
| Sulfur content | ASTM D5453 | mg/L | 600.17 | 467.94 | 303.61 | 151.83 | 9.06 |
| Water content | ASTM D6304 | ppm | 83.28 | 186.66 | 436.74 | 603.47 | 790.20 |

To summarize the results in this section, it was found that biodiesel fuels have great potential as an alternative fuel type for gas diesel engines. This is due to biodiesel properties which are within the acceptable ranges of international standards used in this regard, such as ASTM standard. Thus, biodiesel produced from sheep fats can be used in a diesel engine as diesel fuel substitutes. Moreover, biodiesel produced from sheep fat possess more kinetic viscosity, lubricity and oxidative stability compared to the different blends and diesel fuel, respectively. The density of the biodiesel was also found to be higher than diesel fuel, as expected. These results agree with the findings of Barrios et al. [16] to produce biodiesel from animal fats.

*3.2. Emission Characteristics*

It is also necessary to examine and analyze the performance of the produced sheep fat biodiesel in a diesel engine. Therefore, emissions tests were conducted to see whether the fuel samples can be used efficiently in a diesel engine or not. In this study, exhaust temperature, oxygen, carbon monoxide (CO), carbon dioxide ($CO_2$) and oxides of nitrogen (NOx) emissions (which consist of NO and $NO_2$) were measured. Experimental data for these parameters for sheep fat biodiesel, biodiesel blends and diesel fuel are presented and discussed here.

3.2.1. Temperature of Cooled Exhaust Gases

Temperature of cooled exhaust gases (TCEG) is often related to the ignition delay of a gas diesel. Longer ignition delay results in delayed combustion and increased TCEG. Furthermore, lower cetane number of the fuel prolongs ignition delay. Therefore, TCEG can be increased by combustion, which extends the power period in the cylinder of the engine [17].

The variations of the TCEG values as a function of different load for all fuel types are shown in Figure 9. With the increase in engine load and decreased blend ratio, the exhaust gas temperatures increased. A similar situation was also observed in the study performed by Öztürk [18] for biodiesel and biodiesel blends produced from canola oil. Overall, B20, diesel fuel and B50 shows the highest TCEG of all the tested fuel blends because of their high number of cetane and calorific value. This leads to lower thermal efficiency [19]. Similar high TCEG value for diesel fuel has been reported in the literature [20]. On the other hand, B100 biodiesel fuel produced the lowest TCEG because of its lower cetane number than diesel. An increased in engine load resulted in a corresponding decrease in the oxygen content of all fuel types, as shown in Figure 9a. Additionally, biodiesel

produced from animal waste fat increases the oxygen content (Figure 9b) while decreasing the overall energy content, both of which cause lower TCEG [21].

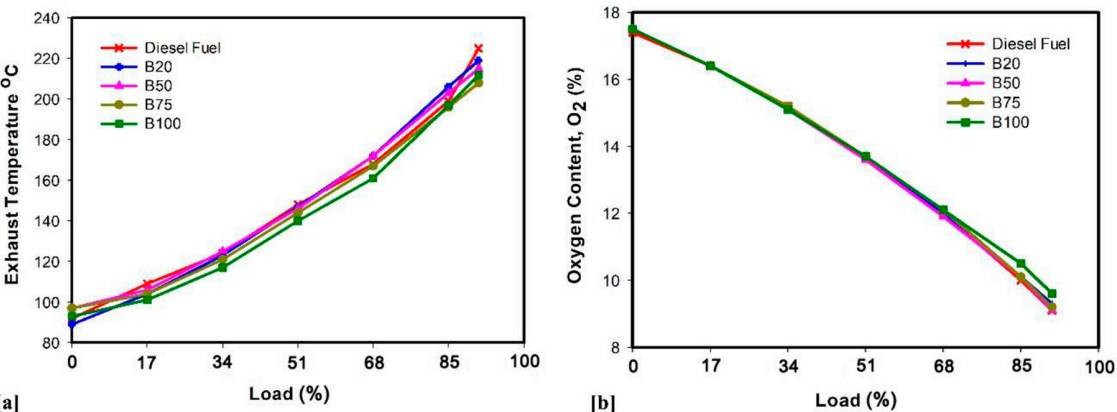

**Figure 9.** (**a**) Exhaust gas temperature and (**b**) oxygen content ($O_2$) as a function of load for diesel, sheep fat biodiesel (B100) and biodiesel blends (B25, B50 and B75) fuels.

### 3.2.2. Carbon Monoxide (CO)

Generally, CO formation results from incomplete combustion [22]. Incomplete combustion occurs when the flame front approaches the crevice volume and a relatively cool cylinder liner [23]. Hence, the flame temperature is cooled down and results in incomplete progression to CO. Comparison between the CO emissions for diesel fuel, sheep fat biodiesel (B100) fuel and biodiesel blends (B20, B50 and B75) fuels is shown in Figure 10. From Figure 10, CO emissions decrease as the load (0–2000 watts) increases for the fuel types. However, as the load was increased from 2000 watts to 3000 watts, a rise in the CO emission level was observed. The increase in CO emissions at higher engine load is because of more fuel needed for higher engine loads, especially for fuel-rich mixtures [24]. The CO of the biodiesel fuels was lower than that of diesel fuel for all load conditions. This is due to the presence of additional oxygen content in biodiesels compared with diesel fuel, which ensures complete combustion. Meanwhile, B20 biodiesel blend exhibited the lowest CO emission at high engine loads. This result is consistent with those reported in the literature [24–26]. For example, Calder et al. [25] found the CO for biodiesel less than the diesel with almost 0.6 g/kWh, with almost 6.6 g/kWh for diesel and 6.0 g/kWh for B20. Di et al. [27] reported that CO emissions increased at low loads while CO emissions decreased at high loads with the addition of ethanol-to-ethanol diesel fuel blends. More so, a B20 blend contains about 35% oxygen and a lower carbon amount in its structure. These properties may also cause the decreased CO emissions of B20 blends at higher engine loads.

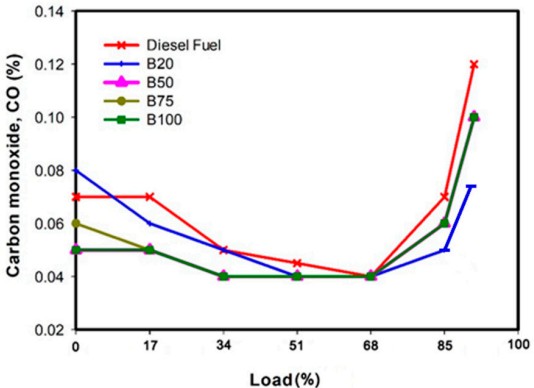

**Figure 10.** Variations of CO with engine load for diesel, sheep fat biodiesel (B100) and biodiesel blends (B20, B50 and B75) fuels.

### 3.2.3. Carbon Dioxide ($CO_2$)

Primary components of exhaust emissions built up using burning hydrocarbon fuels are $CO_2$ emissions. Normally, the air fuel ration in engines significantly affects the making of $CO_2$ and the exhaust amount of it changes by the last of this interaction [27]. Existing maximum level of CO produces decrease in the total $CO_2$ in the atmosphere. If the rate of air-fuel ratio is maintained under control, at an optimal engine load, the rate of CO decreases, and the rate of $CO_2$ rises accordingly. As expected in this study, the $CO_2$ emissions for all fuels increased with the increase in load. The variations of $CO_2$ emissions for the different fuel types are shown in Figure 11. As can be seen from the plot, there was not much difference between the $CO_2$ emissions of biodiesel fuel, biodiesel blends and diesel fuel on average, except B100, which shows a different trend at 34% load compared to all other biodiesel fuels and diesel fuel. This could be explained due to the low load at this point and due to the pure B100 being free from any flammable emission molecules. Similar results were observed for the $CO_2$ emissions of waste chicken fat [24] and waste fish fat [21]. Alptekin et al. (2015) found that compared to fuels, $CO_2$ emissions of diesel and B20 fuels increased by 1.1% and 0.9% on average, respectively. According to the results, there were almost no differences in $CO_2$ emissions of biodiesels and diesel. However, at a low engine load, the $CO_2$ emissions of sheep fat biodiesel were lower than those of diesel fuel. Thus, it was found that $CO_2$ emissions, one of the most important greenhouse gases, can be reduced by using biodiesel at a low engine load.

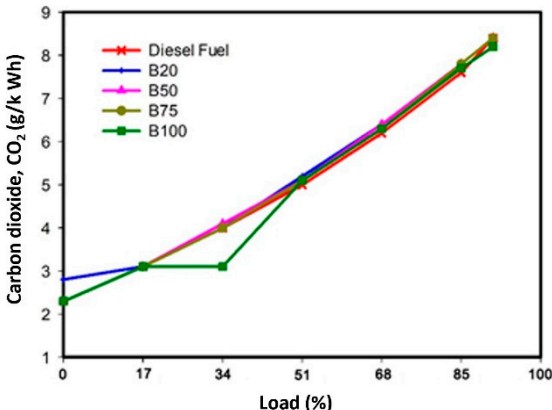

**Figure 11.** Variations of $CO_2$ with engine load for diesel, sheep fat biodiesel (B100) and biodiesel blends (B20, B50 and B75) fuels.

### 3.2.4. Nitrogen Oxide (NOx)

During the combustion process of fuel in the operation of diesel engine, oxygen and nitrogen react with each other at high temperatures and this reaction causes the emission of oxides of nitrogen (NO and $NO_2$), of which NO constitutes a large amount. NOx emission is greatly associated with lean fuel with high cylinder temperature or high peak combustion temperature. A high heat fuel release level at the rapid or premix combustion phase and a decreased heat release level at the mixing/controlled combustion phase will produce NOx emissions [23]. Generally, NO emissions are influenced by the cylinder pressure, temperature and oxygen content of fuel [28]. Figure 12 shows the variations of NO emission values for diesel, sheep fat biodiesel (B100) and biodiesel blends (B20, B50 and B75) fuels. NO emissions increased with increase engine load for all the tested fuels. A higher combustion temperature increases NO by stimulating NO-forming reactions. The NO emissions of B100 were found lower than diesel fuel and biodiesel blends B20, B50 and B75. This is due to their respective chemical constructions; all biodiesels have invariably the same amount of excessive $O_2$ compared with diesel fuel. Moreover, to the inducted air in the engine cylinder, oxygenated biofuels include more $O_2$, which influences the formation of NO. Similar results of other biodiesel produced from animal waste fat

have been reported in the literature [16,21]. Some results from previous studies have been comparable with this study.

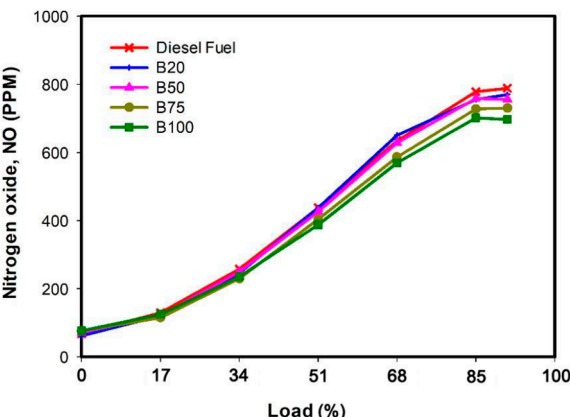

**Figure 12.** Variations of NO with engine load for diesel, sheep fat biodiesel (B100) and biodiesel blends (B20, B50 and B75) fuels.

3.2.5. Nitrogen Dioxide ($NO_2$)

As earlier stated, nitrogen dioxide constitutes a small portion of nitrogen emissions. Figure 13 shows the variations of $NO_2$ with engine load for diesel, sheep fat biodiesel (B100) and biodiesel blends (B20, B50 and B75) fuels. From the Figure 13, it can be seen that emissions of nitrogen dioxide tend to increase slightly with a low load (0–1000 watts) for all fuel types. However, an increase in the engine load from 1000–2700 watts resulted in a significant decrease in the $NO_2$ emission. At high engine loads, B100 resulted in the highest $NO_2$ emissions compared with diesel fuel and the B20, B50 and B75 biodiesel blends. The reason for higher $NO_2$ emissions may be the oxygen content of biodiesel. Alptekin et al. [24] reported that biodiesel from waste chicken fat emitted higher $NO_2$ compared with diesel fuel. A similar result was also observed by [21]. Many researchers have also stated that the reason for the increase in $NO_2$ is the increased cetane numbers of biodiesel which leads to advanced combustion by shortening ignition delay, which promotes $NO_2$ formation reaction [18,21,29]. Overall, NO is one of the greenhouse gases that causes some damages to human health. Thus, NO must be considered and reduced as much possible in the air. NO formation in diesel engines relies on injected fuel kind and quantity, engine fuel, injection time, temperature and the amount of air taken into the cylinder.

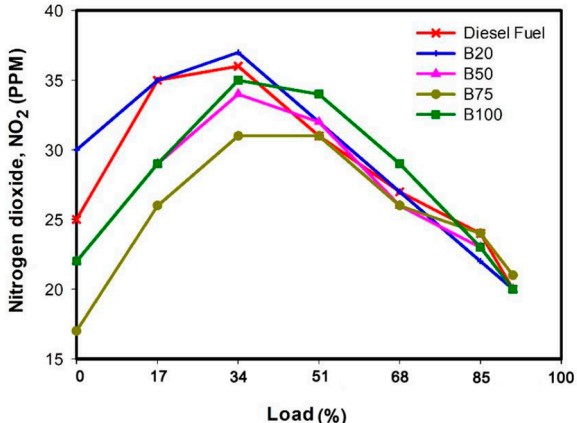

**Figure 13.** Variations of $NO_2$ with engine load for diesel, sheep fat biodiesel (B100) and biodiesel blends (B20, B50 and B75) fuels.

### 3.3. Engine Performance

Figure 14 shows the relationship between the engine load and the engine brake-specific fuel consumption for five kinds of fuel (diesel, B20, B50, B75 and B100). Generally, by increasing the engine load, the brake-specific fuel consumption increases. Each curve represents brake-specific fuel consumption of the five kinds of fuel. This test has been performed for several motor loads (500, 1000, 1500, 2000, 2500 and 2700). The amount of full load torque is 100% maximum load 2700 watts, and by reducing engine load torque, the amount of fuel is reduced. It can also be seen that with the increasing of the load, the variation of BSFC is increased, and the lowest fueling related to load torque is between 7.8 and 8.3. The amount of brake-specific fuel consumption was significantly higher with maximum load torque, and by increasing the load torque, the brake-specific fuel consumption increased. Based on the BSFC curves, B100 was the highest followed by B75, B50, B20 and diesel. This can highlight that the B00 has the highest BSFC and the lowest BSFC is for fossil diesel. Test results showed that brake-specific fuel consumption and volumetric efficiency increased when using blended fuels B20, B50, B75 and B100. Specific fuel consumptions for biodiesel and blends are higher than fossil diesel due to the lower calorific values of biodiesel.

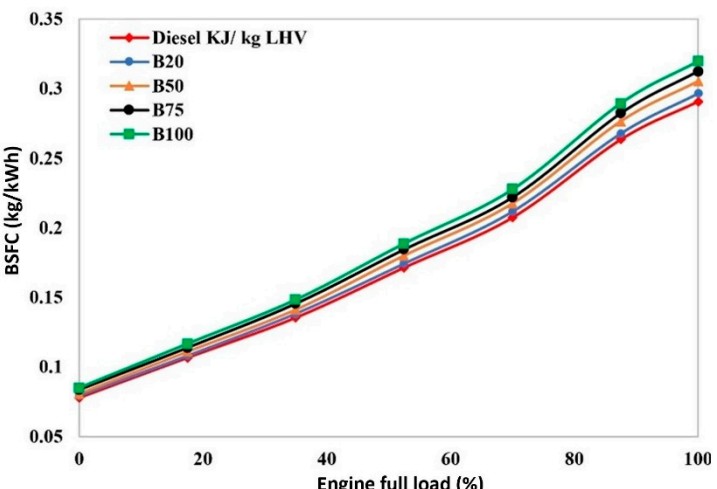

**Figure 14.** BSFC against load for diesel, sheep fat biodiesel (B100) and biodiesel blends (B20, B50 and B75) fuels.

Figure 9a shows the influence of four stroke diesel engine loads on TCEG for fossil diesel, B20, B50, B75 and B100. There is an increase in TCEG with the increase in engine load for all fuels. A decrease in the thermal efficiency of biodiesel and its blends led to increased heat loss in the exhaust gases. This was because of the highest temperature in the engine cylinder, because more fuel is burned to meet the higher load required. The heat loss for exhaust gases was increased with the increase in engine load. As a result of the higher TCEG, poor combustion characteristics are recorded for biodiesel and its blends compared to fossil diesel for the lowest engine load. TCEG values for diesel, B20, B50, B75 and B100 are 93 °C, 88 °C, 98 °C, 98 °C and 93 °C, respectively, at low load. For the entire engine load, exhaust gas temperature values for diesel, B20, B50, B75 and B100 are 225 °C, 220 °C, 215 °C, 208 °C and 212 °C, respectively, at full load.

Brake Mean Effective Pressure (BMEP)

This study also measured the brake mean effective pressure (BMER) of the different fuel type samples (diesel, B10, B20, B30 and B40). This experiment was performed under the range of BMEP from 3.39 bar to 6.35 bar at a rated engine speed range from 1800 rpm to 3600 rpm. The range of BMEP was as the most representative of a wide variety of engine load ranges. Initially, diesel fuel was used as the baseline fuel for the basis of comparison.

Following this, mixtures of biodiesel with 10%, 20%, 30% and 40% volumetric proportions were tested. BSFC shows that the engine performance does not depend on the engine size. From Figure 15, it was observed that the diesel fuel type has the highest amount of BMEP for all the ranges of engine speed except for 2400 rpm, followed by B10, B20 and then by B40. While at the engine speed 3600 rpm, B10 was the highest and B20 was the lowest. This can be attributed to the relatively lower calorific value of the methyl ester fuels being used and, consequently, reduced HRR in the premix combustion region and lower peak combustion temperature [30]. A similar result was observed by Mueller [30], who reported that the higher cetane number of biodiesels blends relative to diesel causes ignition to occur earlier in the cycle. This allows the combustion products to have a longer residence time at high temperatures, which increases NOx emissions. Another possible reason may be associated with the reduction in the heat dissipation by radiation as a consequence of the large reductions in soot emitted with the use of biodiesel.

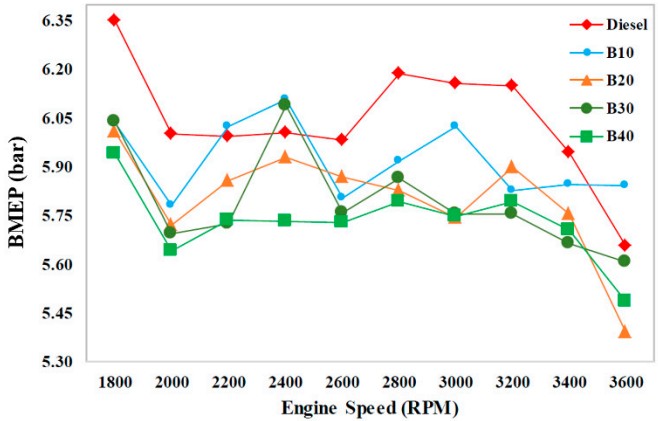

**Figure 15.** The brake mean effective pressure (BMEP) of the different fuel types against the engine speed.

## 4. Conclusions

This research was embarked upon with the aim of creating a sustainable method and system of producing biodiesel from animal fat waste. The research first involved the extraction of sheep fat tallow from sheep. The design of a pilot scale biodiesel plant with a solar power system for powering the reactor was followed. The produced biodiesel was blended in different ratios and comparisons were carried out with the conventional diesel fuels. The combustion and exhaust emission characteristics of sheep fat-based biodiesels, diesel fuel and their blends were determined using a diesel engine. The result of this study was significant and showed that sheep fat can be used as a sustainable source of biodiesel in Kuwait. The choice of a solar power system for powering the transesterification process was justified, as the entire production process was renewable with no use of electricity. The summary of the research findings is presented below, in addition to recommendations for future research. The extraction of sheep fat from sheep was carried out using a simple method and the extraction yield obtained was 80.9% as a biodiesel pure fuel. This biodiesel fuel conformed with the international biodiesel specification; ASTM specifications are listed under ASTM D6751 and the European Organization for Standardization under EN14214 specifying the physical specifications of biodiesel. According to the performance results, the exhaust gas temperature increased from low to high engine loads. The biodiesel (B100) also increased in oxygen content with increased load. CO emissions decreased as the load (0–2000 watts) increased for the fuel types. However, as the load was increased from 2000 watts to 3000 watts, a rise in the CO emission level was observed. There was not much difference between the $CO_2$ emissions of biodiesel fuel, biodiesel blends and diesel fuel on average. At low engine loads, the $CO_2$ emission levels of B100 were lower than those of diesel fuel, B20, B50 and B75. The NO emission of B100 was found to be lower than diesel fuel, B20, B50 and B75. On the other hand, $NO_2$ tended to increase slightly with

low load (0–1000 watts) for all fuel types. However, an increase in the engine load from 1000–2700 watts resulted in a significant decrease in the $NO_2$ emissions. At high engine loads, B100 resulted in the highest $NO_2$ emissions compared with diesel fuel, B20, B50 and B75 biodiesel blends.

**Author Contributions:** Conceptualization, A.A. (Amer Alajmi); methodology, F.A.; validation, A.A. (Ahmed Alrashidi); investigation, F.A. and N.A.; resources, writing—original draft preparation, F.A.; writing—review and editing, N.M.A. and A.A.H. All authors have read and agreed to the published version of the manuscript.

**Funding:** This research received no external funding.

**Institutional Review Board Statement:** Not applicable.

**Informed Consent Statement:** Not applicable.

**Data Availability Statement:** The data presented in this study are openly available in request.

**Acknowledgments:** The authors gratefully appreciate the Public Authority for Applied Education and Training, Kuwait, and Universiti Putra Malaysia (UPM) for providing places and facilitates to complete this project.

**Conflicts of Interest:** The authors declare no conflict of interest.

## Abbreviations

| | |
|---|---|
| B100 | Pure Biodiesel |
| B20 | Diesel + Biodiesel (80:20) |
| B50 | Diesel + Biodiesel (50:50) |
| B75 | Diesel + Biodiesel (25:75) |
| BSFC | Brake-Specific Fuel Consumption |
| BTE | Brake Thermal Efficiency |
| BMEP | Brake Mean Effective Pressure |
| CO | Carbon Monoxide |
| $CO_2$ | Carbon Dioxide |
| TCEG | Temperature of Cooled Exhaust Gases |
| FAAE | Fatty Acid Alkyl Esters |
| FFA | Free Fatty Acide |
| GHG | Greenhouse Gas |
| KOH | Potassium Hydroxide |
| LHV | Lower Heating Value |
| NO | Nitric Oxide |
| $NO_2$ | Nitrogen Dioxide |
| NOx | Nitrogen Oxides |
| $O_2$ | Oxygen |
| PKO | Palm Kernel Oil |
| PV | Photovoltaic |
| Ppr | Power Pulses per Revolution |
| SCE | Supercritical Ethanol |
| SFA | Saturated Fats |
| $SO_2$ | Sulfur Dioxide |
| SPS | Solar Photovoltaic System |

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
