# Peer review of "Development of Solar Powered Biodiesel Reactor for Kuwait Sheep Tallow"

_processes, doi:10.3390/pr9091623_

Round 1

Reviewer 1 Report

The manuscript (processes-1317593) deals with very important problems dealing with technologies for renewable fuels. However, there are several shortcomings that must be taken into consideration and the manuscript must be improved following the remarks:

1) EGT is very low. In fact, this is post-flame temperature, so it is hard to believe the flame temperature was below 100°C and not exceeding 230°C. Hence, it needs verification.

2) Also, methodology for exhaust temperature measurements is welcome.

3) Figures 6a and 12 are the same. It is not reasonable to repeat it.

4) Specifications of the exhaust gas analyzer have to be presented in a separate table showing ranges and accuracies for all measured gases including emissions of NO and NO2.

5) Statement in line 258 is incorrect. The only CO2 emission does not provide any conclusions on combustion completeness.

6) CO2 emission should be expressed in g/kWh rather than percentage scale.

7) Unit for BSFC is incorrect. What is gm?

8) Equations for calculating BSFC, BMEP, and BTE should be provided.

9) Methodologies for measuring air flow rate, fuel flow rate, and engine power with accuracies have to be provided.

10) BTE of 50% is incredibly high. Authors are strongly welcome to describe their methodology for BTE determination, as mentioned in remarks 8 and 9.

11) As regards the PV system, batteries overall capacity is welcome.

The manuscript can be further processed after correcting all the shortcomings.

Author Response

Manuscript Number: 1317593

Reviewer 1

The manuscript (processes-1317593) deals with very important problems dealing with technologies for renewable fuels. However, there are several shortcomings that must be taken into consideration and the manuscript must be improved following the remarks:

1) EGT is very low. In fact, this is post-flame temperature, so it is hard to believe the flame temperature was below 100°C and not exceeding 230°C. Hence, it needs verification.

First, thank you for the reviewer for the fruitful comments. In this study, two thermocouples (-270 °C and 1250 °C) from the temperature of air entering the diesel engine intake and the EGT. Several factors can affect the flame temperature and resulted below 100°C and not exceeding 230°C. The macro jet engine is small in size and allows more clean air to flow into the motor and reset the appropriate air to fuel ratios. These factors such as the small engine used in this study and the particular biodiesel fuel used. For example, B100 biodiesel fuel produced the lowest EGT because of lower cetane number than diesel, and this overall can reduce the EGT. All of these can reduce the EGT in the engine.  EGT values for diesel, B20, B50, B75, B100 are 93°C, 88°C, 98°C, 98°C and 93°C, respectively at low load. for the entire engine load exhaust gas temperature values for diesel, B20, B50, B75, B100 are 225°C, 220°C, 215°C, 208°C and 212°C, respectively at full load. According to the performance results, the exhaust gas temperature increased from low to high engine loads.

2) Also, methodology for exhaust temperature measurements is welcome.

Two thermocouples were used to measure the temperature of air entering the diesel engine intake and the exhaust gas temperature as see in the attached figure below. The thermocouple for intake air temperature was installed directly before the intake port of the throttle body valve, while the thermocouple used to measure the temperature of exhaust gases was installed directly after the exhaust port to avoid any heat losses to the environment. Both thermocouples were k-type with a common range of -270 °C and 1250 °C. Exhaust gas analyser device was used to measure the proportion of combustion exhaust gasses and measures differential flue temperature, carbon dioxide, flue and ambient carbon monoxide, dQa emission of greenhouse gasses. eKH exhaust gas analyser used in this experiment is Eagle x155 with specification shown in Figure 3.15. The combustion specifications are shown in Table 3.6.

3) Figures 6a and 12 are the same. It is not reasonable to repeat it.

This has been corrected by deleting Figure 12.

4) Specifications of the exhaust gas analyzer have to be presented in a separate table showing ranges and accuracies for all measured gases including emissions of NO and NO2.

Specifications of the exhaust gas analyzer has been added in Table 1.

5) Statement in line 258 is incorrect. The only CO2 emission does not provide any conclusions on combustion completeness.

This has been modified.

6) CO2 emission should be expressed in g/kWh rather than percentage scale.

This has been corrected.

7) Unit for BSFC is incorrect. What is gm?

This has been corrected.

8) Equations for calculating BSFC, BMEP, and BTE should be provided.

These have been included.

9) Methodologies for measuring air flow rate, fuel flow rate, and engine power with accuracies have to be provided.

This has been provided in Table 1.

10) BTE of 50% is incredibly high. Authors are strongly welcome to describe their methodology for BTE determination, as mentioned in remarks 8 and 9.

This has been mentioned in section 2.5.

11) As regards the PV system, batteries overall capacity is welcome.

This has been added in section 2.3.

The manuscript can be further processed after correcting all the shortcomings.

Thank you so much for your all-fruitful comments.

Reviewer 2 Report

General comment

  • This study evaluates the potential of using solar energy to produce biodiesel from waste sheep fat. The paper could describe and assess in more detail the impact on CO2 savings just by using solar energy. The paper describes this solar system only marginally so in the context of the publication the solar energy is not essential to the content of the paper. I recommend that the study be extended to include this consideration to assess the impact of solar energy using to CO2 emission as the complex view.
  • A list of abbreviations would be helpful.

Chapter 1. Introduction

  • The literature overview cited in the introduction should be more up-to-date as there are generally 10 years old papers cited.

Chapter 2.1 Experimental Procedures

  • I recommend to describe the chemical reaction of this specific transesterification and to specify sheep fat molecules to be able to provide hypothesises of experimental results.
  • The experimental procedure should be described in much more detail to properly understand Figure 1.

Chapter 2.3 Solar System Design and Calculations

  • Figure 4 is inappropriate with respect to a technical publication. The figures are only demonstrative photographs and do not provide the reader with any valuable information about the technical solution.

Chapter 2.4 Diesel Engine Instrumentations

  • Figure 5 is inappropriate with respect to a technical publication. The figures are only demonstrative photographs and do not provide the reader with any valuable information. There should minimally be introduced and explained the exhaust Gas Analyzer in more detail. Focus should be on measurement accuracy and repeability as well.

Chapter 3.2.2 Carbon Monoxide (CO)

  • The explanation of B20 biodiesel blend CO emission differences at load 85 and 90% seems not to be reasonable and logic. The authors should explain in more detail or repeat the measurement.
  • (Row 251) - This result is consistent with those reported in the literature [24,25]. I recommend to provide these particular results/numbers from reference [24,25]

Chapter 3.2.3 Carbon Dioxide (CO2)

  • The explanation of B100 CO2 emission at the load of 34% should be explained by authors in detail.
  • (Row 267-268) Similar results 267 were observed for the CO2 emission of waste chicken fat [24], and waste fish fat [21]. I recommend to to provide these particular results of waste chicken fat and fish fat from reference [24,21] to compare the data.

Chapter 3.2.4 Nitrogen Oxide (NOx)

  • (Row 267-268) Similar results of other biodiesel produced 292 from animal waste fat have been reported in the literature [16,21]. I recommend to to provide these particular results from reference [16,21] to compare the data.

Author Response

Manuscript Number: 1317593

Reviewer 2

This study evaluates the potential of using solar energy to produce biodiesel from waste sheep fat. The paper could describe and assess in more detail the impact on CO2 savings just by using solar energy. The paper describes this solar system only marginally so in the context of the publication the solar energy is not essential to the content of the paper. I recommend that the study be extended to include this consideration to assess the impact of solar energy using to CO2 emission as the complex view.

  • A list of abbreviations would be helpful.

This has been added before references.

Chapter 1. Introduction

  • The literature overview cited in the introduction should be more up-to-date as there are generally 10 years old papers cited.

Most of the cited previous papers have been updated up-to-date.

Chapter 2.1 Experimental Procedures

  • I recommend to describe the chemical reaction of this specific transesterification and to specify sheep fat molecules to be able to provide hypothesises of experimental results.

This has been included.

  • The experimental procedure should be described in much more detail to properly understand Figure 1.

More details have been added.

Chapter 2.3 Solar System Design and Calculations

  • Figure 4 is inappropriate with respect to a technical publication. The figures are only demonstrative photographs and do not provide the reader with any valuable information about the technical solution.

This has been modified and include more details, figure, and table to explain the measurements methods and accuracies.

Chapter 2.4 Diesel Engine Instrumentations

  • Figure 5 is inappropriate with respect to a technical publication. The figures are only demonstrative photographs and do not provide the reader with any valuable information. There should minimally be introduced and explained the exhaust Gas Analyzer in more detail. Focus should be on measurement accuracy and repeability as well.

This has been modified and include more details and table to explain the measurements methods and accuracies.

Chapter 3.2.2 Carbon Monoxide (CO)

  • The explanation of B20 biodiesel blend CO emission differences at load 85 and 90% seems not to be reasonable and logic. The authors should explain in more detail or repeat the measurement.

This has been corrected by looking back for the raw data and taking the average for all reading instead of taking the last reading Figure 9.

  • (Row 251) - This result is consistent with those reported in the literature [24,25]. I recommend to provide these particular results/numbers from reference [24,25]

Some results from previous studies have been comparable with this study.

Chapter 3.2.3 Carbon Dioxide (CO2)

  • The explanation of B100 CO2 emission at the load of 34% should be explained by authors in detail.

This has been explained.

  • (Row 267-268) Similar results 267 were observed for the CO2 emission of waste chicken fat [24], and waste fish fat [21]. I recommend to to provide these particular results of waste chicken fat and fish fat from reference [24,21] to compare the data.

Some results from previous studies have been comparable with this study.

Chapter 3.2.4 Nitrogen Oxide (NOx)

  • (Row 267-268) Similar results of other biodiesel produced 292 from animal waste fat have been reported in the literature [16,21]. I recommend to to provide these particular results from reference [16,21] to compare the data.

Some results from previous studies have been comparable with this study.

Round 2

Reviewer 1 Report

1) check units: eg. kWh and kWH,

2) check English: eg. shoen(?) and other words

3) data in line 323 does not correspond to any plots.

4) still no information on NOx and NO2 measurement instrumentation.

5) remove the notion "Exhaust Gas Temperature" and EGT.  What you have measured is not exhaust gas temperature. The temperature of exhaust gases is 600-1000C, not 93C.

6) I am afraid, the BTE is wrongly determined. Please provide uncertainty analysis for BTE. Without it, I cannot recommend the manuscript for further processing, sorry, but I have never seen any small engines achieving such a high brake efficiency of 50%.

Author Response

Manuscript ID: 1317593

  • check units: eg. kWh and kWH,

The units have been corrected.

2) check English: eg. shoen(?) and other words

The English in the manuscript has revised.

  • data in line 323 does not correspond to any plots.

We did not correspond the references results in any plots because they did not have similar order, for example they have B0, B5, B20, B50. While we have in this study B0, B20, B50, B75, and B100. So, we just mentioned in general how our results are in agreement with their results.

4) still no information on NOx and NO2 measurement instrumentation.

Information on NOx and NO2 measurement have added in Materials and Methods section as 2.5 Exhaust Emissions Measurement, Lines (231-247).

5) remove the notion "Exhaust Gas Temperature" and EGT.  What you have measured is not exhaust gas temperature. The temperature of exhaust gases is 600-1000C, not 93C.

EGT is relatively low at initial loads, then high of rising engine load. Overall, the engine displacement is slightly small almost 16.11 cubic inch, that’s why the measured EGT in the engine is low.

6) I am afraid, the BTE is wrongly determined. Please provide uncertainty analysis for BTE. Without it, I cannot recommend the manuscript for further processing, sorry, but I have never seen any small engines achieving such a high brake efficiency of 50%.

After an extensive study of previous studies about the BTE, we found that most engines achieving such a high brake efficiency of up to 30-40%. For this, we decided to remove this part from this study and do further investigation about it and study the exact factors if we will find similar efficiency as we found (50%). Then we going to publish the confirmed efficiency results later with uncertainty analysis, thanks to the reviewer.

Round 3

Reviewer 1 Report

The units of BSFC in Fig. 14 are wrong. Not "g/kWh" but kg/kWh.

Unit of pressure (Bar) must be replaced by "bar".

The EGT is still not corrected. Just simply, write it as "temperature of cooled exhaust gases".

Equation for BTE is not required any longer.

Author Response

  • The units of BSFC in Fig. 14 are wrong. Not "g/kWh" but kg/kWh.

This has been fixed.

  • Unit of pressure (Bar) must be replaced by "bar".

This has been fixed.

  • The EGT is still not corrected. Just simply, write it as "temperature of cooled exhaust gases".

The EGT has been corrected in the manuscript.

  • The equation for BTE is not required any longer.

The equation for BTE was removed.
